# Brain White Matter: A Substrate for Resilience and a Substance for Subcortical Small Vessel Disease

**DOI:** 10.3390/brainsci9080193

**Published:** 2019-08-08

**Authors:** Farzaneh A. Sorond, Philip B. Gorelick

**Affiliations:** Davee Department of Neurology, Stroke and Neurocritical Care Division, Northwestern University Feinberg School of Medicine, Chicago, IL 60611, USA

**Keywords:** white matter, small vessel disease, adaptive myelination, oligodendrocytes, oligodendrocyte progenitor cells, cognition, gait, aging, brain magnetic resonance imaging (MRI)

## Abstract

Age-related brain white matter disease is a form of small vessel disease (SVD) that may be associated with lacunar and other small subcortical infarcts, cerebral microbleeds, and perivascular spaces. This common form of cerebrovascular disease may manifest clinically as cognitive impairment of varying degrees and difficulty with mobility. Whereas some persons show cognitive decline and mobility failure when there are brain white matter hyperintensities (WMH) and acute stroke, others recover, and not everyone with brain white matter disease is disabled. Thus, repair or compensation of brain white matter may be possible, and furthermore, certain vascular risks, such as raised blood pressure, are targets for prevention of white matter disease or are administered to reduce the burden of such disease. Vascular risk modification may be useful, but alone may not be sufficient to prevent white matter disease progression. In this chapter, we specifically focus on WMH of vascular origin and explore white matter development, plasticity, and enduring processes of myelination across the health span in the context of experimental and human data, and compare and contrast resilient brain white matter propensity to a diseased white matter state. We conclude with thoughts on novel ways one might study white matter resilience, and predict future healthy cognitive and functional outcomes.

## 1. Introduction

Human brain white matter is a substantial expanse of cerebral tissue that acts as a support structure for the transfer of critical signals between the cortex, subcortex, and more distal central nervous system neuroanatomic structures. This bidirectional informational transfer highway serves to assure proper signaling of vital messages and disposal of toxic substances during healthy function. It is subject, however, to the ravages of small vessel disease when there is vascular dysfunction, and disruption of the blood-brain barrier. Although cerebral white matter is vulnerable to injury and dysfunction, it has the capacitance for developmental and adaptive myelination and repair. The latter functions form a possible epicenter for protection of the brain, and provide mechanistic pathways for heightening brain health.

In this review, we discuss the underpinnings of unhealthy or dysfunctional brain white matter and associated clinical manifestations of such injury, and then transition to an understanding of how primate white matter develops, sustains itself, and has adaptive and plastic properties to allow healthy survival. By better understanding and harnessing the potential of cerebral white matter’s developmental and adaptive mechanisms, a new avenue for maintenance of brain health may become manifest.

## 2. White Matter Hyperintensities (WMHs) Are Associated with Vascular Risk Factors (VRFs)

In the three decades since Hachinski et al. [1] coined the term “leukoaraiosis” (LA), investigations on the clinical and pathological correlates of cerebral white matter changes have increased dramatically. In the neuro-vascular world, age-related periventricular and deep hypointenisties on CT (LA) or hyperintensities on MRI (white matter hyperintensities: WMHs), are now considered one of the most prevalent surrogate neuroimaging expressions of subcortical small vessel disease (SVD) (PMID: [2] (Figure 1A,B)). These radiographic changes, which are strongly associated with vascular risk factors (VRFs) such as arterial hypertension and history of stroke, also demonstrate neuropathological evidence of arteriolar lipohyalinosis, venous collagenosis. and capillary rarefactions [3] as the underlying causes [4,5,6].

## 3. WMHs Are Also Associated with Aging and They Are Not Innocuous

WMHs have also been associated with age-related cognitive and mobility impairment, and as such, are no longer considered innocuous findings. More than 90% of the general population older than 80 years of age has some degree of WMHs [7], and it is well established that WMHs lead to cognitive decline and contribute to dementia and functional disability. However, what is strikingly clear is that these associations are not straightforward. And in some instances, the sizes of the effects are rather small. Furthermore, persons with PIB+AD (Pittsburgh compound B positive Alzheimer’s Disease) may have greater WMH volume, and those with amyloid burden in cerebral amyloid angiopathy (CAA) may have higher amounts of WMH [8,9]. Thus, WMHs may track with amyloid burden and contribute to chronic cerebral ischemia.

## 4. WMHs and Cognition

A number of meta-analyses of existing cross-sectional and longitudinal studies have attempted to quantify the effect of WMH volume and progression on cognitive function in the general population and in those with diagnosis of Alzheimer’s disease (AD). A meta-analysis of nine prospective longitudinal studies published between 1996–2009 that used MRI to assess the impact of WMHs on the risk of incident dementia shows that WMHs were associated with a near two-fold increased risk of dementia (HR 1.9, [1.3 to 2.8]) [10]. More recently, meta-analysis of studies conducted between 1990 and 2013 examined the quantitative effect of WMH progression and localization on specific cognitive domains [11]. In this analysis, the presence of WMHs was significantly associated with cognitive deficits in all domains, but the overall effect size was small (r −0.1 [−0.13, −0.08]). Progression of WMHs was also associated with greater cognitive decline across all domains, but decline was most pronounced in attention and executive functioning. The authors suggested that the absence of domain-specific cognitive deficits attributed to WMHs is evidence that WMH effects on cognition in the general population differ from the neurodegenerative changes associated with specific memory deficits seen in AD. More importantly, the authors also argued that their findings should prompt us to ask not whether WMHs have clinical relevance, but rather, which persons have clinically significant effects from WMHs.

In a more recent study, this work has been further extended to examine the clinical relevance of WMHs in patients with AD and mild cognitive impairment (MCI) [12]. The premise for this investigation was studies that show WMHs are associated with increased risk of AD [10,13] and modify the cognitive profile of AD [14,15,16,17,18]. Moreover, WMHs seem to even be a core feature in autosomal dominant AD [19]. A few studies have also suggested that WMH may predict MCI conversion to AD [20], as well as accelerating cognitive decline in patients with AD [21]. So in another meta-analysis using data from 12 studies on AD and 10 studies on MCI, these investigators showed that WMHs were significantly associated with overall cognition in both AD and MCI. Interestingly, while WMHs had an effect size in the AD group that was similar to the general population, (−0.11 [−0.14, −0.08]), the impact of WMHs was significantly stronger in the MCI group (−0.25, [−0.36, −0.14]), with the most robust effects on attention, executive functions, and processing speed. In contrast, a meta-analysis by Liu et al.; reported that AD pathology was significantly associated with WMHs in AD patients, but not in healthy age-matched controls [22].

## 5. WMH and Gait

WMHs are also commonly associated with slow gait and impaired mobility. The Leukoaraiosis And Disability (LADIS) study, which has significantly contributed to the existing knowledge of WMH and cognition, has also advanced our understanding of WMHs and disability, particularly in the context of age-related functional decline [23,24]. The LADIS study has shown that the severity of WMHs is a strong and independent predictor of the transition from an autonomous status to disability. In the context of gait and mobility, data from the LADIS study show that WMHs are also related to physical performance (Short Physical Performance Battery, SPPB) and rate of falls [25,26]. But, similar to WMHs and cognition, studies involving mobility and WMHs are also challenged by the degree to which WMHs account for the observed gait impairments [27,28,29,30,31]. Clinical experience also supports results from existing cross-sectional and longitudinal studies of WMHs in relation to mobility and cognitive outcomes, where we often see individuals with substantial burden of WMHs who have “normal” psychomotor processing speed, “normal” physical function, “normal” detailed cognitive testing, and higher than expected mean gait speed.

## 6. WMHs Are Age and Vascular Risk Factor-Related and Associated with Cognitive and Mobility Impairment, Yet the Effect Size Is Variable and Not Everyone Is Affected

In summary, accumulated evidence has established LA or WMHs are common in older adults, and associated with cognitive and gait dysfunction. Controlling VRFs should reduce the burden of SVD and white matter injury, but preventive strategies have not proven sufficient. Moreover, the association between WMHs, cognition, and mobility is not straightforward, and in some instances, the effect sizes are rather small. This is in line with clinical observations that while some people with these white matter changes are inexplicably afflicted with mobility and cognitive impairment, others are strikingly resilient and are able to maintain their cognitive health and functional independence. Therefore, in addition to controlling VRFs, we need to also understand what mechanisms distinguish those individuals with a preponderance of WMHs who remain cognitively and functionally independent from those who do not. Elucidation of these mechanisms will allow the examination of the biological underpinnings of resilience in small vessel disease, and target these pathways towards preserving brain health across the life or health span.

## 7. White Matter Microstructure, Vascular Risk Factors, Cognition, and Mobility

Accumulated evidence from newer imaging techniques, such as diffusion tensor imaging (DTI), suggests that WMHs represent the extreme end of a continuous spectrum of white matter injury. Such measures that can detect microstructural changes in white matter tract integrity before they can be seen as WMHs on standard neuroimaging studies [32,33] show that the health of the white matter microstructure may be a source of resilience, and that individuals with a healthy white matter microstructure may be able to tolerate a much higher burden of the macrostructural white matter changes seen as WMHs.

DTI, which measures directionality of proton (water) diffusion on a microscopic spatial scale, generates three measures of diffusivity: axial (Ad), along the direction of axons; radial (Rd), perpendicular to Ad; the summary diffusivity measure mean diffusivity (MD), the average of Adand Rd, as well as another summary variable, fractional anisotropy (FA). FA is thought to reflect fiber density, axonal diameter, and myelination. In tightly organized fiber bundles, diffusion is restricted to the long axis of the fibers and anisotropy is high, whereas in the newborn white matter, diffusion is unrestricted and anisotropy is low [34]. As the fiber tracts mature and myelination proceeds, diffusion declines and anisotropy (FA) increases. There is growing evidence that alterations reflected in and measurable with diffusion imaging continue throughout the course of brain development and degeneration. As human white matter matures, denser packing of axons that results from myelination, along with increases in axonal diameter, are most likely limiting the unrestricted water in extra-axonal space [35].

Evidence linking VRFs to DTI measures and WM microstructural integrity is also accumulating and robust (for recent reviews see [34,35]). In one of the largest and most comprehensive analyses to date, investigators from the Atherosclerosis Risk in Communities Neurocognitive [36] study support prior work and demonstrate that hypertension and diabetes were both associated with worse white matter microstructural integrity according to DTI measures. Most importantly, they also showed that associations between VRFs and white matter microstructural integrity were largely independent of WMH volumes (consistent with prior reports, [37,38,39]), reinforcing the notion that DTI measures provide an assessment of pathologic changes that precede and predict the development of WMHs or white matter loss [36,40].

Similarly, a growing body of evidence also links cognition and mobility with white matter integrity detected by DTI [41,42,43]. These studies show that microstructural changes in normal-appearing white matter predict faster decline in psychomotor speed, executive functions, and working memory, regardless of WMH burden [24,44]. In 340 participants from the LADIS study, DTI changes within normal-appearing white matter increased with WMH severity, and unlike DTI measures within WMH, normal-appearing white matter changes had a strong and independent effect on cognitive functions [45]. Along the same lines, in a population-based sample of 860 community-based dementia-free participants from the Netherlands, the microstructural integrity of both WMHs and normal-appearing white matter was associated with cognitive function (information processing speed, global cognition, and memory), regardless of white matter atrophy and WMH volume [46].

In a cohort of 265 community dwelling older adults from the Health ABC study, the association between WMHs and gait speed was moderated by FA, such that in those with lower FA, the WMH-gait association was stronger than in those with higher FA [47]. In a smaller study of 86 participants with longitudinal data, regional tract-based analysis showed that a decrease in FA and increase in RD within the normal-appearing white matter genu of the corpus callosum correlated with slower walk time independent of age, gender, and WMH burden over a 4-year follow-up [48]. Similarly, in 493 participants from the Baltimore Longitudinal Study of Aging, data show that in the younger elderly, tract-based microstructural white matter changes in FA and MD in normal-appearing white matter are associated with gait changes independent of WMHs, suggesting that disrupted regional white matter microstructure may represent an earlier phase of white matter injury and age-related mobility impairment [49]. Similarly, in 275 participants from the Radboud University Nijmegen Diffusion Tensor and Magnetic Resonance Imaging Cohort (RUNDMC) with longitudinal gait and imaging data, decreased white matter integrity in several white tracts was associated with decline in gait, with the strongest associations in the corpus callosum and corona radiata, which were also independent of WMHs [50].

In summary, DTI has had a significant impact on our understanding of white matter injury in SVD. We now know that in individuals with small vessel disease, normal-appearing white matter deteriorates with the presence of increasing WMHs and age, confirming that normal-appearing white matter in the presence of even a few WMHs is not ‘normal’ [51,52]. In fact, WMHs are likely the tip of the iceberg in small vessel disease, and their impact on age related cognitive and mobility decline should be considered in context of the overall health of the white matter microstructure. In other words, individuals with a healthy white matter microstructure (DTI measures) may be able to tolerate a much higher burden of macrostructural white matter changes (WMHs). Clearly, a more precise understanding of the white matter changes associated with SVD and white matter health as a substrate for vulnerability and/or resilience in the aging brain are needed. Knowledge in this domain will be a critical step towards developing novel targets for interventions to promoting healthy brain aging and reducing the burden of dementia and functional dependence in our aging population.

## 8. Brain White Matter: Developmental Aspects and Components

The human brain is unique in its high content of myelin, the fatty sheath that coats our nerve axons and constitutes brain white matter [53,54,55]. Transmission of nerve impulses and the fast processing speeds essential to the integration and performance of our complex cognitive abilities hinge on myelin development, maturation, degeneration, and repair—all mechanisms that continue into adulthood [56,57,58,59,60]. Computational studies have shown that unlike the gray matter volume, which as a percentage of total brain volume is similar across all anthropoid primates, relative white matter volume increases with brain size from 9% in pygmy marmosets to about 35–45% in humans, the highest value in primates [61,62,63]. The hyper scaling of white matter volume relative to brain size may be mainly driven by the higher rate of change in white matter (neural connections) than in gray matter (neural elements) relative to the brain size [62].

With the advent and widespread use of MRI for noninvasive and safe characterization of brain macro- and microstructure and function in vivo, much earlier computation work can now be validated using imaging. These studies also support a greater proportion of white matter volume in humans [54,55,64,65]. Studies characterizing human neurodevelopmental trajectories, both in health and disease [56,66,67,68,69,70], have significantly advanced our understanding of developmental and degenerative changes in the human brain. These imaging studies have demonstrated that white and gray matter trajectories follow different patterns during development. Based on absolute and normalized volumes, white matter trajectories follow an inverted U-shape, with a maturation peak around mid-life and a growth rate that is faster during maturation than volume decrease during aging. On the other hand, while there are some exceptions for subcortical gray structures (e.g.; the amygdala and hippocampus) overall, gray matter volumes appear to increase between 1 to 10 years of age and then decrease. However, since initial gray matter growth is related to body growth, when normalized volumes are considered, gray matter volumes continuously decrease across the life span.

*Susceptibility and Vulnerability of White Matter*. Developmentally, white matter fiber tracts develop at variable rates. While some are fully developed at birth and by early childhood, others continue to develop through adolescence and early adulthood [71,72,73,74,75,76]. The dynamic pattern of white matter development is also reflected in the dynamic pattern of WMH development. Therefore, the timing of brain imaging should be an important consideration in assigning regional preponderance of WMH burden across the life span [77]. Moreover, the pathological underpinning of WMH significantly contributes to the dynamic pattern of WMH progression [78]. Nonetheless, from a development perspective, the brain areas and connections with the most protracted changes are located in the prefrontal and temporal cortices, and demonstrate significant heterogeneity across trajectories and regions [66]. While early myelinating fibers are heavily myelinated and robust, these later-myelinating fibers have a smaller diameter and are more vulnerable to normal aging and/or degenerative processes [79]. This concept, which has been termed “retrogenesis” [80,81], is based on the observation that later myelinating regions (e.g.; frontal subcortical white matter tracts) are more susceptible to WMH and DTI changes in SVD, as well as the pattern of amyloid deposition. The pathological correlates of the ischemic WMH in these areas comprise several patterns, including myelin pallor or swelling, tissue rarefaction associated with loss of oligodendrocytes, diffuse axonal injury with thinning and varicosities, loosening of axon-oligodendrocyte adhesion, and gliosis. Given consistent observations of the importance of white matter in normal and abnormal cognitive development and aging, identification of the factors that increase susceptibility to white matter degeneration remains an important avenue of research.

Most of the CNS myelin content (50%) is in white matter in humans, but compared to other species, our gray matter is also highly myelinated [82], suggesting that inter-species disparity of myelin content, rather than the white matter content, may be the differentiating factor for human brain development. However, the high metabolic demand of myelin may also be the source of vulnerability to aging and small vessel disease in the human brain [83]. Moreover, the presence of myelin in gray matter suggests that the impact of myelin pathology is not limited to white matter, and that we should broaden our view of the clinical consequences of these disorders.

*Adaptation and Plasticity of Myelin*. Myelination in the central nervous system results from spiral wrapping or ensheathment of axons by the plasma membranes of oligodendrocytes separated by segments of compact myelin to form internodes. Oligodendrocytes, which are derived from oligodendrocyte precursor cells (OPCs), continue to proliferate and differentiate into oligodendrocytes well into adulthood (for detailed review, see [84]). OPCs continuously survey their environment and are capable of migrating to areas of demand to replace injured or dying cells [84] and even reinitiate myelin growth in the adult by wrapping mature myelin sheaths with additional layers of membrane [85].

Moreover, there is a rapidly expanding literature indicating that myelination is also a dynamic and adaptive process (adaptive myelination; [84,86,87]), that continues throughout life and shapes the innate myelin infrastructure patterned during development. Myelin plasticity clearly complements established neuronal processes, such as long-term potentiation and neurogenesis, that are linked to cognitive processes. Adaptive myelination is defined by the continued myelin alterations into adulthood that result in response to experience and neuronal activity and shape the cellular and microstructural properties of myelin necessary to modify network dynamics and behavior (for detailed review, see [58,88]). This dynamic feedback loop between myelin plasticity and neuronal activity is crucial for the elaboration and stabilization of neuronal circuitry, helps strengthen motor and cognitive function, and permits the acquisition of new skills and memories in children and adults.

The accompanying Table 1 lists interventions and briefly summarizes results of interventional studies designed to modify white matter. The studies are largely based on primary outcomes related to changes in FA according to a specific intervention (e.g.; juggling, playing music, learning a new language, etc.). Thus, one can see in Table 1 that the various activities may lead to favorable changes in FA, and that myelin may be adaptively altered in a beneficial manner.

In addition to the cross talk between neuronal activity (neurons) and adaptive myelination (oligodendrocytes), cerebral endothelial cells and OPCs are also coupled, wherein endothelial-derived trophic factors regulate OPC behavior [103]. In response to injury, OPC respond with extensive proliferation, migration, and morphological changes in an attempt to regenerate myelin. Endothelial cell-derived trophic factors include FGF-2, BDNF and VEGF A. In turn, OPC-derived TGF-β maintains the blood-brain barrier (BBB) integrity through development, thus supporting the cerebrovascular system. However, conditions such as hypoxic-ischemic injury can disrupt OPC-endothelium trophic coupling and promote the progression of white matter diseases. Without the trophic support from the endothelial cells, myelin repair fails and injury is extended to involve BBB injury through OPC-derived MMP-9 secretion (for a detailed review, please see [104]). Remyelination failure results from two different mechanisms. First, OPC regulation is downregulated, resulting in increased OPC proliferation but decreased differentiation [105]. Second, glutamate release resulting neuronal excitotoxicity directly hinders remyelination [106]. Within brain white matter, which is most vulnerable to small vessel injury, myelinated axons, oligodendrocytes, OPC, astrocytes, and microglia that are embedded in the extracellular matrix and nourished by a network of capillaries extending from the perforating or medullary end arteries provide an “oligovascular” niche [103,107,108,109] that regulates the health of the white matter. So, considering that tissue regeneration requires not only progenitor cells, but also functional vascular systems, the health of the “oligovascular” niche is crucial for maintaining white matter integrity. Identifying mechanisms to maintain the health of the “oligovascular” niche and enhance adaptive myelination in response to injury may be the best complement to VRF modification in preserving brain health across the life and health span.

It is also well established that normal brain development is contingent upon critical genetic signaling and essential environmental input. In fact, through sequential intrinsic and adaptive pathways, myelination creates a “smart wiring” that changes alongside synaptic plasticity, and enables learning as well as preserving brain health [110]. The smart wiring relies on myelination within an initially hard-wired pattern (intrinsic myelination during development) that can undergo adaptation to modify myelin sheath number and/or properties [86] in response to experience (adaptive myelination) and enable CNS plasticity and learning. Hence, the developmental hard wiring is as critical to brain health as the adaptive myelination for cognitive function across the life span (see Table 1). So, while in the context of brain aging and vascular injury we have focused on adaptive myelination as substrate for preserving brain health, the influence of intrinsic myelination as a critical contributor to resilience cannot be underestimated.

## 9. Concluding Thoughts: The Role for Developmental and Adaptive Myelination in Brain Health

The brain and heart are intimately linked by a central vascular highway that provides a flow of essential nutrients and the disposal of metabolic waste products, a continuum of environmental exposures and intrinsic factors shaping the internal milieu responsible for vital brain functions. This vascular network is intimately connected in the brain to a glial network integral to neural circuitry, from astroglial vascular end-feet to peri-synaptic processes and encompasses neurovascular as well as “oligovascular” niches that are at the core of brain health and function. Since cognitive decline typically manifests in mid-life or later, often times we overlook the role of fetal and developmental origins of brain health (e.g.; intrinsic myelination) as a target for intervention. Could the hard wiring of the brain (intrinsic myelination) be a source of reserve when the “oligovascular” niche becomes ischemic and white matter microstructure (DTI) and macrostructure (WMHs) injury ensue? Does intrinsic myelination determine the adaptability of the regenerating myelin? Are those with less robust intrinsic myelination (environment and genetic contributions) more vulnerable to cognitive and mobility decline in SVD?

Identifying methods to recruit the mechanisms of adaptive myelination in the context of disease, which offer the potential to drive ongoing myelination in the healthy adult brain, are of great interest in the realm of regenerative medicine. Adaptive changes in myelin-forming cells represent a type of behaviorally-relevant brain plasticity, raising numerous conceptual and mechanistic questions. Mechanisms regulating myelin plasticity may be important for adaptive neural function and could be leveraged for interventions in diseases of myelin. However, just like controlling VRFs is not sufficient to preserve brain health, only focusing on adaptive myelination without regard for intrinsic myelination would also be short sighted. In fact, the life course perspective and recommendations outlined in the Presidential Advisory statement by the AHA/ASA on Defining Optimal Brain Health in Adults [111] specifically addresses the essential multifactorial approach necessary to preserve brain health. Their recommendations specifically target development (intrinsic myelination) and maintenance (adaptive myelination) of “smart wiring” through education, good nutrition, exercise, VRF control, meaningful interpersonal and community social engagement, and access to mental health care. Taken together, these findings emphasize early-life determinants of later-life cognitive function and illustrate the underappreciated plastic nature of myelin in the mammalian brain. While numerous conceptual and mechanistic questions remain, the pervasive and protracted process of myelination in the human brain may hold promise for novel therapeutic approaches towards maintaining brain health across the life and health span.

## Figures and Tables

**Figure 1 brainsci-09-00193-f001:**
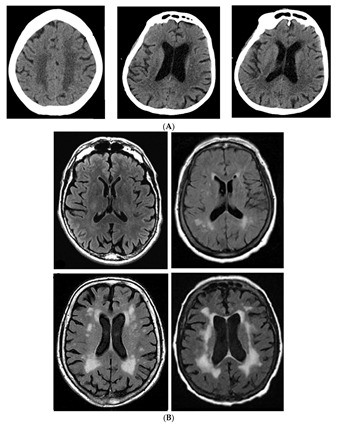
(**A**) Non-contrast brain computed tomography with leukoaraiosis. (**B**) FLAIR MRI sequences with increasing severity of deep and periventricular white matter hyperintensities.

**Table 1 brainsci-09-00193-t001:** Studies in Activity Dependent Myelination.

InIntervention	Finding	Study
**Experimental Models**		
Social isolation: 4 weeks of isolation in juvenile mice	Deficits in social interaction task not rescued by social re-introduction mPFC: reduction in ramification of mature astrocytes, reduced internode per oligo, thinning of myelin sheath	[89,90]
Complex wheel task in adult mice	Additional OPCs and mature oligodendrocytes in the corpus callosum. Generation of new oligodendrocytes required skill improvement	[91]
Rats trained in single-pallet reaching task	Increase FA in Cingulum and external capsule	[92]
Optical stimulation of the pre-motor cortex in mice	Proliferation of OPCs in the deep cortex and subcortical white matter within the stimulated circuit. Activity induced expansion of the oligodendrocyte lineage	[93]
**Human Studies**		
Juggling training in adults	Increased FA	[94]
Musical training	Increased FA	[95]
Learning a second language	Increased FA	[96]
Extensive piano practicing	Enhanced white matter development	[97]
finger-thumb opposition sequence task (10 min daily, over 4 weeks)	Increased FA	[98]
Visuomotor skill training	Increased myelination only in ROIs contralateral to trained limb, correlated with skill acquisition (Increased myelin water fraction [MWF])	[99]
Exercise (Physical fitness or Activity: PFA)	Increased PFA with improved WM structure, but effect size small	[100]
Memory Training	Those with higher MD had the least improvement	[101]
Cardiorespiratory fitness (exercise training)	Cardiorespiratory fitness (exercise training)	[102]

mPFC: medial Prefrontal Cortex; OPCs: Oligodendrocyte Progenitor Cells; FA: Fractional Anisotropy; ROIs: Regions of Interest; MD: Mean Diffusivity.

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
