# Peer review of "Brain White Matter: A Substrate for Resilience and a Substance for Subcortical Small Vessel Disease"

_brainsci, 2019, doi:10.3390/brainsci9080193_

Round 1
Reviewer 1 Report
Interesting review highlighting the regenerative capacity of the cerebral white matter in light of its involvement in small vessel disease.
Author Response
Thank you, we appreciate this reviewers input. Will complete another spell check
Reviewer 2 Report
The perspective article by Sorond and Gorelick provides an excellent overview on the links between white matter disease and small vessel disease. The paper provides a good list of literature references for further study and the authors have indicated this very well throughout the paper. There are some important changes that would strengthen the paper.
Major
1. I want to ask the authors to expand the section on the oligovascular niche, specifically on how these cells interact and through which mediators. I think this should be at least mentioned in a review on white matter and small vessel disease. Hamanaka et al describe these mediators such as MMP-9, TGFb1 and VEGF among others.
2. It would be very nice to summarize the review in a figure for readers who prefer to visualize. The authors could think of a figure in which WMH and LA are shown with the corresponding resulting impairment in cognitive function and mobility (some data from table 1) and show the link with small vessel disease and the oligovascular niche (with the mediators of the preceding point). I realize that this is not a small task but it would help a lot of readers.
3. Line 151: is it possible to report regression data between WM microstructural integrity on DTI and functionality in the same way as reported in the earlier paragraph on the association of WMH on MRI with cognitive / mobility function?
4. Table 1: separate the animal and the human studies in the table, I would advise to add a column indicating whether animals (which one) or humans were studied; also, under the table, the abbreviations should be written out
5. The references need to be reviewed: author names need to be separated by a comma and I would advice to truncate the number of authors after the third name by adding et al.
Minor
1. Line 34: is this manuscript a chapter?
2. Line 61: the abbreviation PIB+AD has not been written out
3. Line 93: I would replace “normal” with “healthy” or “controls without AD”
4. Line 111: the abbreviation VRF (vascular risk factor) has not been written out – the authors could add the abbreviation to the title of the paragraph
5. Line 186: I would also add this first sentence to the introduction of the manuscript since a lot has already been said on white matter
6. Line 210: typo “e”
7. Line 257: typo “additional” should be “addition”
Author Response
Please see uploaded document

Reviewer 3 Report
This is a lucid and interesting article.
One concept alluded to but not addressed directly in the manuscript is the possible heterogeneous nature of WMH's. Are all WMH's that appear on MRI reflective of the same pathological processes or not. The data presented would suggest otherwise as do some pathological studies. This is reflective in the statement that begins on line 103. Also the dynamic pattern of WMH's is not really developed in the discussion. Are there any insights into the posterior preponderance of some WMH's that often are seen earlier than more frontal 'WMH's. In this regard are there vascular territory (posterior vs anterior) or other susceptibilities at play? Does disease process have an effect ie AD vs CVD?
Brain Microbleeds are mentioned in the abstract but not discussed in the text. Brain microbleeds have been related to the extent of WMH's but what is know about the relation to DTI's?"
One of the most interesting parts of the manuscript is the discussion regarding the usefulness of DTI and the information gained by this method regarding the development of white matter injury. The concept that WMH is that tip of the iceberg in compelling but the statement on line 122 may have to be modified or explained a bit since as pointed out an individual may have a high burden of WMH on MR but essentially normal DTI parameters. The authors suggest that it is the DTI parameters are the most important and this point could have even more emphasis.
Perhaps the role of the BBB and BBB opening as an injury mechanism could be mentioned as per the work of Drs BV Zlokovic at USC and G Rosenberg at U NM.
Some vascular anatomy is mentioned (line 267) but more information could be made regarding differences in blood flow and perfusion between gray and white matter and the effects of aging on such - again contributing to selective susceptibility as developed in part in the manuscript. This is alluded to in the section around lines 268 and thereabouts and the concept of the "oligovascular niche" is very interesting and could be explained in more detail as the reference (no 87) is not all that helpful.
The sentence on line 224 needs to be clarified or the number quoted is wrong; "most of the CNS myelin content is in the white matter (50%) ...Does that mean the 50% exists elsewhere in the CNS? Would that not mean that the distribution is equal? Please clarify.
Author Response
Please see uploaded document
